

# The Parenting to Reduce Adolescent Depression and Anxiety Scale: Assessing parental concordance with parenting guidelines for the prevention of adolescent depression and anxiety disorders

Mairead C. Cardamone-Breen[1], Anthony F. Jorm[2], Katherine A. Lawrence[1], Andrew J. Mackinnon[2,3] and Marie B.H. Yap[1,2]

[1] School of Psychological Sciences, Monash Institute of Cognitive and Clinical Neurosciences, Monash University, Melbourne, Victoria, Australia
[2] Melbourne School of Population and Global Health, University of Melbourne, Melbourne, Victoria, Australia
[3] Black Dog Institute, University of New South Wales, Sydney, New South Wales, Australia

Corresponding author
Marie B.H. Yap,
marie.yap@monash.edu

## ABSTRACT

**Background**. Despite substantial evidence demonstrating numerous parental risk and protective factors for the development of adolescent depression and anxiety disorders, there is currently no single measure that assesses these parenting factors. To address this gap, we developed the *Parenting to Reduce Adolescent Depression and Anxiety Scale (PRADAS)* as a criterion-referenced measure of parental concordance with a set of evidence-based parenting guidelines for the prevention of adolescent depression and anxiety disorders. In this paper, we used a sample of Australian parents of adolescents to: (1) validate the PRADAS as a criterion-referenced measure; (2) examine parental concordance with the guidelines in the sample; and (3) examine correlates of parental concordance with the guidelines.

**Methods**. Seven hundred eleven parents completed the PRADAS, as well as two established parenting measures, and parent-report measures of adolescent depression and anxiety symptoms. Six hundred sixty adolescent participants (aged 12–15) also completed the symptom measures. Concordance with the guidelines was assessed via nine subscale scores and a total score. Reliability of the scores was assessed with an estimate of the agreement coefficient, as well as 1-month test-retest reliability. Convergent validity was examined via correlations between the scale and two established parenting measures.

**Results**. One proposed subscale was removed from the final version of the scale, resulting in a total of eight subscales. Reliability was high for the total score, and acceptable to high for seven of the eight subscales. One-month test-retest reliability was acceptable to high for the total score. Convergent validity was supported by moderate to high correlations with two established measures of parenting. Overall, rates of parental concordance with the guidelines were low in our sample. Higher scores were associated with being female and higher levels of parental education. Greater parental concordance with the guidelines was associated with fewer symptoms of depression and anxiety in adolescent participants.

**Discussion**. This initial validation study provides preliminary support for the reliability and validity of the PRADAS. The scale has potential for use in both clinical and research settings. It may be used to identify parents' strengths and potential targets for intervention, and as an outcome measure in studies of preventive parenting interventions.

## INTRODUCTION

Depression and anxiety disorders are among the largest contributors to global burden of disease, and are leading causes of disability in young people (*Kessler et al., 2007*; *Mathews et al., 2011*; *Patel et al., 2007*). These disorders have peak onset early in life, with approximately half of all cases emerging by age 14, and 75% by 24 years (*Kessler et al., 2005*). Early onset of depression and anxiety is associated with a cascade of negative long-term sequelae (e.g., *Copeland et al., 2009*; *Dekker et al., 2007*; *Woodward & Fergusson, 2001*), leading to recognition that prevention of these disorders early in life is a global healthcare priority (*Patel et al., 2007*).

Existing research has demonstrated numerous risk and protective factors for the development of depression and anxiety early in life, many of which are shared between the disorders. The overlap of risk factors and high degree of comorbidity has led to the recommendation of a transdiagnostic preventive approach (*Garber, 2006*). Many risk factors are not modifiable (e.g., female gender, genetic disposition; *Garber, 2006*; *Mathew et al., 2011*; *Rapee, Schniering & Hudson, 2009*; *Van Voorhees et al., 2008*), or are difficult to intervene with at an individual level (e.g., socioeconomic status, family history of mental illness; *Garber, 2006*; *Mathew et al., 2011*; *Van Voorhees et al., 2008*). However, a number of identified factors involve the family environment, or can be detected and responded to by parents. For example, inter-parental conflict, over-involvement, parental aversiveness, and lack of family support have been shown to increase risk (*Hankin, 2006*; *Rapee, 2012*; *Rapee, Schniering & Hudson, 2009*; *Van Voorhees et al., 2008*; *Yap et al., 2014a*); whereas parental warmth, family connectedness, and autonomy granting reduce risk (*Van Voorhees et al., 2008*; *Yap et al., 2014a*). Thus, targeting specific risk and protective factors that are within parents' control is likely to be effective in preventing depression and anxiety in young people. Consideration of family and parenting factors in the prevention of depression and anxiety has been recognised as a key research translation priority (*Avenevoli & Merikangas, 2006*; *Fisak Jr, Richard & Mann, 2011*; *Restifo & Bögels, 2009*). Promisingly, a growing body of evidence supports the efficacy of preventive parenting interventions for reducing risk of mental illness in children and adolescents (e.g., *Sandler et al., 2011*; *Siegenthaler, Munder & Egger, 2012*; *Yap et al., 2016*).

To translate research evidence related to the role of parenting in the prevention of adolescent depression and anxiety, a set of guidelines were developed for parents

of adolescents: "How to Prevent Depression and Clinical Anxiety in Your Teenager: Strategies for Parents" (henceforth *the Guidelines*; *Parenting Strategies Program, 2013*; http://www.parentingstrategies.net/depression). The Guidelines were developed through a two-phase process involving: (1) a systematic review of parental risk and protective factors for adolescent depression and anxiety (*Yap et al., 2014a*); and (2) a Delphi expert consensus study to establish parenting strategies that are important for the prevention of depression and anxiety disorders in adolescents (*Yap et al., 2014b*). The Guidelines include 190 practical parenting strategies, organised under 11 topics: you can reduce your child's risk of depression and anxiety problems; establish and maintain a good relationship with your teenager; be involved and support increasing autonomy; establish family rules and consequences; minimise conflict in the home; encourage supportive relationships; encourage good health habits; help your teenager deal with problems; help your teenager to deal with anxiety; encourage professional help-seeking when needed; and don't blame yourself.

Despite the large body of research demonstrating the importance of parenting factors, there is currently no single measure that specifically assesses the range of parenting factors shown to influence the development of depression and anxiety in adolescents. Some factors, such warmth, control, and aversiveness, have been extensively researched using widely-used measures (e.g., the Parental Bonding Instrument, *Parker, Tupling & Brown, 1979*; the EMBU [Egna Minnem av Barndoms Uppfostram], *Perris et al., 1980*; the Children's Report of Parental Behavior Inventory, *Schaefer, 1965*). However, other factors (e.g., encouraging sociability, encouraging good health habits) have limited research and no validated measures. In order to effectively target the range of parenting risk and protective factors in preventive interventions, a valid measure to assess these factors is needed. To address this gap, the current paper describes the development and validation of the *Parenting to Reduce Adolescent Depression and Anxiety Scale (PRADAS)*—a parent-report measure of parenting practices against the recommendations in the Guidelines. As the Guidelines recommend specific, defined strategies, we developed the scale as a *criterion-referenced* measure. In contrast to commonly used *norm-referenced* tests, in which an individual's performance is compared with that of others, criterion-referenced tests assess knowledge or skills against defined domains of content (*Hambleton & Rogers, 1990*). These tests are used to determine whether an individual has reached a pre-defined level of competence or mastery of a skill (*Hambleton & Rogers, 1990*). Norm-referenced tests are unable to provide this information, hence are not appropriate when this is the purpose of assessment. Based on the research informing the Guidelines, higher levels of parental concordance with the Guidelines are expected to be protective against adolescent depression and anxiety disorders.

## Aims and hypotheses

In this paper, we used a sample of Australian parents of adolescents aged 12–15 to: (1) validate the newly developed PRADAS as a criterion-referenced measure of parental concordance with the Guidelines; (2) examine levels of parental concordance with the Guidelines; and (3) examine correlates of parental concordance with the Guidelines. We examined the reliability and validity of the scale to fulfil the first aim. Reliability

was assessed via an estimate of the *agreement coefficient*, as well as 1-month test-retest reliability. Validity of the scale was determined based on its face validity and convergent validity indicators. Convergent validity was assessed via correlations between the scale and two previously validated measures of parenting: the Inventory of Parent and Peer Attachment (IPPA) and the Emotions as a Child scale (EAC). As these measures each have some overlapping content with our scale, we predicted at least moderate correlations. We predicted positive correlations with the subscales assessing positive parenting practices, and negative correlations with the subscales assessing undesirable parenting practices. We also expected small to moderate correlations between the nine PRADAS subscales and moderate to high correlations between the subscale scores and the total score. Finally, we explored the correlates of parental Guidelines concordance with participant characteristics (e.g., demographics, mental health history, adolescent depression and anxiety symptoms). We predicted small correlations between parental Guidelines concordance and symptoms of depression and anxiety in adolescent participants. Examination of other correlates were exploratory in nature, hence no hypotheses were specified.

## METHODS

### Participants and procedures

Participants were parent-adolescent dyads participating in one of two randomised controlled trials (RCTs) of online parenting interventions aimed at preventing adolescent depression and anxiety disorders. Eligible participants were parents or primary caregivers of an adolescent aged 12 to 15 years, who resided in Australia and had regular internet access. No exclusion criteria were specified. Parent participation was not dependent on adolescent participation; as such, parents could participate without their child. Only one parent-adolescent dyad per family was eligible to participate. Recruitment was via online parenting networks, social media, university networks, mental health organisations, and schools across Australia. Ethics approval was obtained from the Monash University Human Research Ethics Committee (approval numbers CF14/3886–2014002023 and CF14/3887–2014002024).

Parent participants self-selected by responding to advertisements and registering and providing consent for themselves and their adolescent via the trial websites. Parents and adolescents were provided with participant explanatory statements, detailing participation requirements, privacy, and standard limits to confidentiality (e.g., disclosure of risk of harm to self or others, child abuse). A member of the research team called adolescent participants to explain the study and obtain their verbal assent. Adolescents who agreed to participate were guided through their online baseline assessment over the phone, which included the child-report version of the Spence Children's Anxiety Scale (SCAS) and Short Mood and Feelings Questionnaire (SMFQ). Assistance to complete the measures was provided by the researcher where necessary (e.g., in the case of learning disabilities). Adolescents were reimbursed for their participation with an e-voucher. Following submission of the adolescent baseline assessment, an automated email was sent to the parent participant containing the link to their baseline assessment. Parent participants completed the PRADAS, and the parent-report versions of the IPPA, EAC, SCAS, and SMFQ.

The two RCTs from which participants were drawn assessed similar interventions (i.e., a brief and intensive version of the same online parenting program) and targeted similar populations. Further, data used in the current paper was collected prior to any intervention being received. It was therefore deemed acceptable to combine baseline data for the purposes of this paper. Demographic characteristics of participants from the two samples were compared to assess for significant differences on key variables. There were no significant differences between samples for the following variables: parent or child age; parent or child gender; parent employment status; parent marital status; parent–child relationship (i.e., mother/father/other); parent- or child- report scores on the SCAS or SMFQ; or total score on the PRADAS (all $ps > .05$). Significant differences were found for three demographic characteristics: (1) percentage of parents speaking a language other than English (8.6% in sample 1 vs. 15.8% in sample 2, $\chi^2[1, N = 711] = 8.62, p = .003$); (2) highest parent qualification, with a greater number of participants in sample 1 having post-graduate qualifications (32.9% vs. 24.9%) and more parents in sample 2 having secondary school (year 7 to 12) qualifications (13.9% vs. 7.7%), $\chi^2[5, N = 711] = 15.75$, $p = .008$; and (3) state of residence, with more participants in sample 2 residing in Victoria (86.1% vs. 21.4%, $\chi^2[7, N = 711] = 316.63, p < .001$). As these variables were not outcomes, nor expected to significantly affect results, we opted to combine the samples in order to attain a larger and more diverse sample of parents in Australia.

A total of 350 and 361 parents registered and completed baseline assessments for the two trials respectively. The final sample therefore comprised 711 parents. Most parents were female (89.5%), with a mean age of 45.15 years ($SD = 5.66$). Six hundred sixty adolescents also consented to participate and completed baseline assessments. Adolescents had a mean age of 13.66 years ($SD = 1.05$) and 47.3% were female. Table 1 presents additional participant demographic characteristics.

## Measures
### The Parenting to Reduce Adolescent Depression and Anxiety Scale (PRADAS)
As discussed, the PRADAS was developed as a criterion-referenced measure of parental concordance with the Guidelines. The scale therefore assesses current parenting practices against the highest-quality evidence for parental risk and protective factors for adolescent depression and anxiety. The scale also forms part of two newly developed individually-tailored online parenting interventions which are currently being evaluated in two RCTs (trial registrations available at: https://www.anzctr.org.au/Trial/Registration/TrialReview.aspx?ACTRN=12615000247572, https://www.anzctr.org.au/Trial/Registration/TrialReview.aspx?id=368274).

Item selection and writing are critical to the validity of criterion-referenced tests (*Hambleton & Rogers, 1990*). The consensus of experts on the topic is one suggested way of ensuring item validity (*Hambleton & Rogers, 1990*). The items were developed and comprehensively revised during multiple workshops by authors MCB, MY, AJ, and KL. Two of the authors (MY and AJ) were authors of the Guidelines, as well as a number of systematic reviews on the role of parenting in the prevention of child and adolescent depression and

**Table 1   Descriptive statistics for parent and child demographic characteristics.**

| Parent/child characteristic | N | % |
|---|---|---|
| **Parent relationship to child** | | |
| Mother | 631 | 88.7 |
| Father | 73 | 10.3 |
| Grandmother | 2 | 0.3 |
| Step-mother | 2 | 0.3 |
| Step-father | 1 | 0.1 |
| Aunt | 1 | 0.1 |
| Guardian | 1 | 0.1 |
| **Parent relationship status** | | |
| Married/de facto | 548 | 77.1 |
| Separated/divorced | 120 | 16.9 |
| Single | 37 | 5.2 |
| Widowed | 6 | 0.8 |
| **Family situation** | | |
| Intact family, child living with both parents | 486 | 68.5 |
| Separated parents, shared care | 71 | 10.0 |
| Child living with one parent (participant) | 132 | 18.6 |
| Child living with one parent (not participant) | 9 | 1.3 |
| Other | 12 | 1.7 |
| **Parent employment status** | | |
| Working full time | 319 | 44.9 |
| Working part time | 309 | 43.5 |
| Unemployed | 83 | 11.7 |
| Studying (full or part time) | 133 | 18.7 |
| **Parent education level** | | |
| Secondary school year 7 to 12 | 77 | 10.8 |
| Trade/apprenticeship | 7 | 1.0 |
| TAFE certificate/other technical qualification | 70 | 9.8 |
| Diploma | 126 | 17.7 |
| Bachelor degree | 226 | 31.8 |
| Postgraduate | 205 | 28.8 |
| Language other than English spoken at home | 87 | 12.2 |
| **State of residence** | | |
| Victoria | 386 | 54.3 |
| New South Wales | 120 | 16.9 |
| Queensland | 78 | 11.0 |
| Tasmania | 54 | 7.6 |
| South Australia | 25 | 3.5 |
| Australian Capital Territory | 25 | 3.5 |
| Western Australia | 22 | 3.1 |
| Northern Territory | 1 | 0.1 |

**Notes.**
TAFE, Technical and Further Education.

anxiety (e.g., *Yap et al., 2016*; *Yap & Jorm, 2015*; *Yap et al., 2014a*). These authors also have experience developing a similar criterion-referenced measure of parenting practices for the prevention of adolescent alcohol misuse (*Yap et al., 2011*; *Yap, Jorm & Lubman, 2015*). The remaining authors were a graduate clinical psychology student conducting doctoral research on the topic (MCB), and an experienced clinical psychologist and academic with expertise in the field (KL).

The original version of the scale included 79 items, across nine subscales (6–12 items each), corresponding to the sections of the Guidelines. The first and last sections of the Guidelines were not included in the scale, as they provide psychoeducation rather than specific parenting recommendations. Table 2 presents a summary of the PRADAS subscales, Guideline topics, and risk/protective factors assessed in each. Most questions relate to a specific parenting behaviour recommended in the Guidelines (e.g., eating dinner together as a family), assessed on a Likert-type frequency scale (e.g., never, rarely, sometimes, often). Other questions relate to hypothetical situations (e.g., noticing a persistent change in the adolescent's behaviour), and enquire how likely a parent is to take certain actions (e.g., very unlikely, unlikely, likely, very likely). In an attempt to reduce response bias, twenty-six items were negatively worded or false-positive questions (e.g., the opposite behaviour to the recommendation in the Guidelines). File S1 contains the scale items.

*Face validity.* The scale was designed to have face validity, based on the opinion of the developing authors. This was confirmed via consultation with a reference group of parents of adolescents (see below).

*Consultation with target-end users.* To ensure acceptability of the measure to target-end users, we consulted a reference group of 22 parents (19 mothers, 3 fathers) of adolescents aged 11 to 18 years. Reference group parents were recruited via staff newsletters at Monash University and the University of Melbourne, local school newsletters, and social media. Parents attended one of three repeated focus group workshops ($n = 7$ to 8 per workshop), during which they were shown a draft of the scale. Parents provided feedback about the presentation, acceptability, wording, and order of items. Where possible, this feedback was incorporated into the final version of the scale.

*Scoring of the scale for Guidelines concordance.* As the scale is a criterion-referenced measure, a cut-off score for mastery of each item and subscale was carefully selected by the developing authors. Responses for each item were deemed either concordant or non-concordant, based on the content of the Guidelines. Depending on the specific recommendation, some items were given only one concordant response (e.g., for the false positive item "When [my teenager] is upset, I encourage [him/her] to toughen up", only "never" was considered concordant). However, we allowed more "lenient" scoring for items that may differ between families or situations, but are still considered concordant with the Guidelines' recommendation (e.g., for the item "I make time to ask [my teenager] about [his/her] day and what [he/she] has been doing", answers of "often" or "sometimes" were considered concordant). Concordant item responses were scored as 1 and non-concordant responses as 0. We summed the item scores to yield nine subscale scores and a total score.

**Table 2**  PRADAS subscales, guidelines topics covered, number of items, risk/protective factor assessed, and example items.

| PRADAS subscale | Guidelines topic covered | No. of items | Risk/protective factors covered | Example items[a] |
|---|---|---|---|---|
| Parent-child relationship | Establish and maintain a good relationship with your teenager | 8 | Parental warmth, aversiveness, affection, emotional availability | I let my [my teenager] know that I love [him/her]. I can tell when [my teenager] is open to talking with me. |
| Involvement | Be involved and support increasing autonomy | 8 | Parental over-involvement, autonomy granting, monitoring | I increase [my teenager]'s responsibilities and independence over time (e.g., let [him/her] make more decision about [his/her] life). |
| Relationships with others | Encourage supportive relationships | 6 | Parental encouragement of sociability | I encourage [my teenager] to spend time with [his/her] friends. |
| Family rules | Establish family rules and consequences | 9 | Consistency of discipline | Have you set specific, defined rules for [your teenager]'s behaviour? Was [your teenager] involved in developing the family rules? |
| Home environment | Minimise conflict in the home | 8 | Inter-parental conflict, parent-child conflict management, criticism, parental modelling of conflict management | If I argue with my partner, I make sure that [my teenager] can't hear. When I have an argument or conflict with [my teenager], I problem solve the issue with [him/her]. |
| Health habits | Encourage good health habits | 12 | Diet, physical activity, sleep hygiene (7 items); responding to alcohol or drug use (5 items) | I have good health habits (i.e., healthy diet, regular exercise, responsible use of alcohol) myself. I allow my teenager to have an alcoholic drink at home to help [him/her] learn to drink responsibly. *(False positive).* |
| Dealing with problems | Help your teenager to deal with problems | 10 | Problem solving, emotion regulation, stress management, modeling of problem solving approaches | I encourage [my teenager] to work towards realistic goals. I give up on tasks that prove to be too difficult. *(False positive).* |
| Coping with anxiety | Help your teenager to deal with anxiety | 9 | Anxiety management (avoidance, exposure), modeling of anxiety management strategies | If my teenager takes steps to manage [his/her] anxiety, I praise [him/her] for doing it. |
| Professional help-seeking | Encourage professional help-seeking when needed | 9 | Professional help-seeking knowledge and behaviours (parent and child) | If you noticed a persistent change in [your teenager]'s mood or behaviour, how likely would you be to: encourage [him/her] to talk to you about what's going on. Take [him/her] to a trained mental health professional. |

Notes.

[a] Square brackets in items denote personalisation with the adolescent's name and gender.

The total score can therefore range from 0 to 79. Subscale scores depend on the number of items in the subscale (see Table 2). A cut-off score was also set to indicate concordance for each subscale (i.e., mastery of the skill area). These cut-off scores were intentionally set to be high, so as to indicate close to absolute concordance with Guidelines. We set this as our criterion for mastery, as the Guidelines are based on high quality research evidence

and international expert consensus. The recommendations are therefore the ultimate goal which parents are encouraged to achieve. Concordance was set as one score less than 100% for all subscales except for the *health habits* subscale, which is the longest subscale (12 items). This subscale has seven items on general health habits (diet, physical activity, sleep hygiene) and five items on alcohol or drug use. We allowed a total of two non-concordant responses in this subscale (cut-off score of 10).

In addition to the PRADAS items, we collected parent-reported information on parent and child demographic characteristics, parent and child history of mental illness, and parental concern about their child's risk of developing depression and anxiety. Other measures used for cross-validation of the PRADAS are described below.

### Inventory of parent and peer attachment—parent-report (IPPA-P; *Armsden & Greenberg, 1987*)

The IPPA was originally developed as an adolescent-report measure of attachment to parents, measuring three cognitive-affective dimensions of attachment: mutual trust, quality of communication, and alienation in relationships. We administered an adapted 25-item parent-report version of the scale (*McElhaney et al., 2008*; JP Allen, pers. comm., 2013) to parents. Items were scored on a 5-point Likert scale (never true, seldom true, sometimes true, often true, almost always true). We calculated the three subscale scores (trust, communication, and alienation) as well as a total score. The original version of the inventory has demonstrated good internal consistency reliability (Cronbach's alpha .89 to .92) and three-week test-retest reliability (.93; *Greenberg & Armsden, 2009*), as well as convergent validity with other scales of family functioning and environment (*Armsden & Greenberg, 1987*; *Gullone & Robinson, 2005*). The adapted version used in the current study has high correlations with the original version (.92 to .98), and acceptable to high Cronbach's alphas (.70 to .89 for total score, trust and communication subscales; .68 to .70 for the alienation subscale; JP Allen, pers. comm., 2013). A similar parent-report version has also been found to have high internal consistency (total score Cronbach's alpha = .92; *McElhaney et al., 2008*). Cronbach's alphas for our sample were high for the total score (.90), acceptable for the trust (.87) and communication (.79) subscales, and just below acceptable for the alienation subscale (.69).

### Emotions as a child scale (EAC; *Magai, 1990*, as cited in *O'Neal & Magai, 2005*)

The EAC is a 45-item parent self-report inventory that assesses parental response to children's expression of sadness, anger, and fear, according to five emotion socialisation strategies: support (e.g., comforting or assisting the child); override (e.g., minimising the emotion); magnify (e.g., amplifying the emotion); neglect (e.g., not responding); and punish (e.g., critical or punishing responses). We calculated both the emotion-specific and global subscale scores, yielding a total of 15 emotion-specific subscales and five global emotion socialisation subscales. The EAC has previously demonstrated acceptable to good internal consistency reliability (*Garside & Klimes-Dougan, 2002*; *Klimes-Dougan et al., 2007*; *O'Neal & Magai, 2005*; *Silk et al., 2011*) and acceptable one-week test-retest reliability (*Klimes-Dougan, Brand & Garside, 2001*, as cited in *Garside & Klimes-Dougan, 2002*;

*O'Neal & Magai, 2005*). Internal consistency reliability for the current sample was acceptable to high for the global emotion socialisation subscales (Cronbach's alphas: reward = .88; punish = .81; neglect = .79; override = .86; magnify = .84). Six of the 15 emotion-specific subscales had Cronbach's alphas greater than .70, but the remaining nine ranged from .55 to .68. We therefore opted to utilise the global emotion socialisation subscales in our analyses.

### Spence children's anxiety scale (SCAS; *Spence, 1997*)

The SCAS is a 45-item child-report (SCAS-C) and 39-item parent-report (SCAS-P) measure that assesses six domains of anxiety: separation anxiety, social phobia, obsessive-compulsive disorder, panic/agoraphobia, generalised anxiety, and fear of physical injury. Respondents rate the degree to which the child experiences a number of symptoms on a 4-point frequency scale (never, sometimes, often, always). Six of the child-report items are non-scored filler items designed to reduce response bias. Items are scored from 0 (never) to 3 (always), and summed to form a total anxiety score. The SCAS total score has demonstrated high internal consistency reliability (Cronbach's alpha = .93) and adequate test-retest reliability between three and six months (.63 and .60, respectively) in children aged 8 to 14 years (*Spence, 1997*; *Spence, 1998*; *Spence, Barrett & Turner, 2003*). The SCAS has also been normed on Australian school children aged 8 to 15 years, similar to the current sample (*Spence, 1998*; *Spence, Barrett & Turner, 2003*). We calculated the total score for parent- and child-report versions, both of which had high internal consistency in our sample (Cronbach's alphas: parent-report = .91; child-report = .94).

### Short mood and feelings questionnaire (SMFQ; *Angold et al., 1995*)

The SMFQ is a 13-item child- (SMFQ-C) and parent-report (SMFQ-P) scale assessing depressive symptoms in children and adolescents. The scale is a shortened version of the original 32-item Mood and Feelings Questionnaire (*Angold et al., 1995*). Respondents rate each item on a three-point scale of 0 (not true), 1 (sometimes true) or 2 (true) to indicate the frequency of depressive symptoms in the previous two weeks. The scale has been used widely in children aged 6 to 17 years, across various samples including psychiatric, paediatric, and community populations (e.g., *Angold et al., 1995*; *Messer et al., 1995*; *Rhew et al., 2010*; *Thapar & McGuffin, 1998*). The SMFQ has been shown to correlate highly with other widely-used measures of depression in children and with structured clinical interviews (*Angold et al., 1995*; *Kovacs, 1984*). The scale has also shown high internal consistency reliability (Cronbach's alphas = .85 for child-report, .87 for parent-report), and good sensitivity and specificity for detecting cases of depression (*Angold et al., 1995*). Cronbach's alphas for our sample were high (.90 for parent-report; .91 for child-report).

### Data analysis

Proportion of missing data was low at both the subject and item level. Less than 5% of cases had missing data on any particular measure. All items on the PRADAS were mandatory due to the programming of the online parenting interventions, therefore all parents answered all items. Due to a technical error, four items had missing data for between one and three participants. This was deemed to be missing completely at random and replaced using

Expectation Maximisation (EM) imputation. Overall proportion of missing data was less than 1% for all other measures. Missing data for the IPPA, EAC, SCAS and SMFQ were replaced using EM for cases with less than 30% of data missing on any measure. Participants with missing data greater than 30% on a measure were excluded from analyses.

Concordance with the Guidelines was initially examined with descriptive statistics for the total and subscale scores. Frequencies were calculated for parent and child characteristics.

As the scale is a criterion-referenced measure, conventional internal consistency reliability indices are not considered appropriate (*Berk, 1980*; *Hambleton & Rogers, 1990*; *Subkoviak, 1988*). Alternative reliability indices for criterion-referenced tests include the *agreement coefficient* and the *kappa coefficient* (*Hambleton & Rogers, 1990*; *Subkoviak, 1988*). The agreement coefficient is a measure of the proportion of examinees consistently classified on two or more administrations of a test; whereas the kappa coefficient reflects the gain in consistency provided by using the test over an unreliable measure (*Subkoviak, 1988*). We chose to utilise the agreement coefficient, as our purpose was to assess the overall consistency of the measure in classifying parents as concordant or non-concordant with the Guidelines. Subkoviak suggested that agreement coefficients of 0.75 or greater are acceptable in situations where half of test-takers are masters and half are non-masters. Agreement coefficients increase as the proportion of masters to non-masters becomes more dissimilar (*Subkoviak, 1988*). We utilised the tables provided by Subkoviak to calculate an estimate of the agreement coefficient based on a single administration of the measure, for the subscale scores and the total score.

Test-retest reliability was assessed by calculating the correlation between baseline and 1-month follow-up assessment scores for a subset of participants ($N = 175$) in the waitlist control group of one of the RCTs.

Validation of conventional norm-referenced tests typically involves structural modeling techniques such as exploratory or confirmatory factor analysis to assess the extent to which items represent underlying latent constructs. In contrast, criterion-referenced tests may assess a broad range of skills within a given domain, all deemed necessary to achieve "mastery" of the domain. The individual items comprising each domain may not be expected to reflect a single factor, as items may represent distinct skills. As such, factor analyses of items may not be appropriate. In the case of the current scale, the skills covered within each domain of the Guidelines were not expected to uniformly represent nine distinct constructs or factors. Instead, we considered the items to represent a set of skills that were deemed necessary to be concordant within a particular parenting domain. As such, we did not employ conventional factor analytic techniques. Instead, we were interested in the interrelationships between the domains and overall concordance with the Guidelines. We examined the correlation matrix of subscale scores and the total score. We predicted small to moderate correlations between the subscales, as each subscale was designed to assess distinct parenting skills. As we expected concordance within one domain to be associated with overall concordance, we predicted moderate to high correlations between the subscale scores and total score.

To assess convergent validity of the scale, we examined correlations between the total and subscale scores of our measure and those of the EAC and IPPA. Finally, we examined

bivariate correlations between parent and child characteristics (gender, age, education, mental health history, scores on SCAS and SMFQ) and overall Guidelines concordance. We also conducted exploratory analyses to assess the ability of the PRADAS to distinguish between participants above and below clinical cut-offs on the SCAS and SMFQ, using logistic regressions.

## RESULTS

### Parent and child mental health characteristics

Table 3 presents frequencies for parent and child history of mental health problems and parental concern about their adolescent's risk of developing depression or anxiety. Most parents expressed at least "a little" concern about their adolescent's risk of developing depression (79.9%) or anxiety (79.0%). Over half (58.5%) of parent participants reported experiencing either past or current mental health problems themselves. Regarding adolescent mental health history, parents reported that 15.1% of adolescents had been diagnosed with a mental health problem in the past, and 22.1% believed that their child had experienced a mental health or behavioural problem that had not been formally diagnosed. Similarly, 17.2% of adolescents were reported to be experiencing current mental health issues, with 24.1% of parents concerned that their child may be currently experiencing an undiagnosed mental health or behavioural problem.

### Concordance with the Guidelines

Table 4 presents descriptive concordance statistics for the nine subscales and the total score. Concordance rates for most subscales were low, with less than 20% of parents scoring within the concordant range for seven of the nine subscales. This may in part be due to the fact that cut-off scores for concordance were intentionally set to be high. Two subscales were notable exceptions to this pattern: (1) *Relationships with others* had a much higher concordance rate of 77.5%. The mean score of this subscale was close to the highest possible score (5.11 out of 6.00) and there was less variance in responses, representing a ceiling effect; (2) *Professional help-seeking*, which also had a higher concordance rate than other subscales, at 49.37%.

Further exploration of items within the *relationships with others* subscale revealed ceiling effects for four of the six items (94% to 99% of participants concordant), explaining the overall ceiling effect. These items assessed parental encouragement of socialisation, being kind to others, and talking to the adolescent about social problems (see File S1 for scale items). The remaining two items in this subscale, which were both negatively worded, had concordance rates of 70.2% and 53.2%.

Examination of the *professional help-seeking* subscale revealed ceiling effects for three of the nine items. Two of these items assessed parental response to the question "If you noticed a persistent change in your teenager's mood or behaviour, how likely would you be to: (insert question)". Ninety-nine percent of parents responded that they would encourage their teenager to speak to them about what was going on, and 98.7% endorsed that they would try to determine whether the change was caused by a temporary situation or more ongoing problem. Further, 93% of parents stated that they would know where to seek

**Table 3** Frequencies for parental concern about adolescent risk of depression and anxiety, and parent and child history of mental health problems.

| Parent/child characteristic | N | % |
|---|---|---|
| Parental concern about child's risk of developing depression | | |
| Not at all | 139 | 19.5 |
| A little | 338 | 47.5 |
| Yes | 150 | 21.1 |
| Very much so | 80 | 11.3 |
| Missing (declined to answer) | 4 | 0.6 |
| Parental concern about child's risk of developing an anxiety disorder | | |
| Not at all | 145 | 20.4 |
| A little | 323 | 45.4 |
| Yes | 151 | 21.1 |
| Very much so | 89 | 12.5 |
| Missing (declined to answer) | 3 | 0.4 |
| Parental history or current mental health problem | | |
| None | 290 | 40.8 |
| Yes, past history | 277 | 39.0 |
| Yes, current | 75 | 10.5 |
| Yes, past and current | 64 | 9.0 |
| Missing (declined to answer) | 5 | 0.7 |
| Child history of mental health or behavioural disorder diagnosis[a] | | |
| None | 425 | 59.8 |
| Depression | 15 | 2.1 |
| Any anxiety disorder | 50 | 7.0 |
| Other mental health or behavioural disorder | 56 | 7.9 |
| No formal diagnosis, however I believe my child has experienced some emotional or behavioural problems | 157 | 22.1 |
| Missing (declined to answer) | 34 | 4.8 |
| Child current mental health or behavioural problems[a] | | |
| None | 437 | 61.5 |
| Depression | 24 | 3.4 |
| Any anxiety disorder | 79 | 11.1 |
| Other mental health or behavioural disorder | 55 | 7.7 |
| No formal diagnosis, however I believe my child is currently experiencing some emotional or behavioural problems | 171 | 24.1 |
| Missing (declined to answer) | 13 | 1.8 |

**Notes.**

'*Other mental health or behavioural disorder*' category includes: Autism Spectrum Disorder, Attention-Deficit/Hyperactivity Disorder, Oppositional Defiant Disorder, Conduct Disorder, learning difficulties, or any other disorder specified by parents.
[a] Percentages add to >100% as multiple responses were allowed.

**Table 4** Descriptive and reliability statistics for PRADAS subscales and total score (N = 711).

| PRADAS subscale | Highest possible score[a] | Cut-off score for concordance | Observed minimum | Observed maximum | M | SD | % concordant | Agreement coefficient |
|---|---|---|---|---|---|---|---|---|
| Parent-child relationship | 8.00 | 7.00 | 0.00 | 8.00 | 5.00 | 1.27 | 11.67 | .84 |
| Involvement | 8.00 | 7.00 | 1.00 | 8.00 | 5.26 | 1.28 | 17.86 | .77 |
| Relationships with others | 6.00 | 5.00 | 2.00 | 6.00 | 5.11 | 0.80 | 77.50 | .68 |
| Family rules | 9.00 | 8.00 | 1.00 | 9.00 | 4.57 | 1.69 | 4.36 | .98 |
| Home environment | 8.00 | 7.00 | 0.00 | 8.00 | 4.23 | 1.67 | 9.85 | .89 |
| Health habits | 12.00 | 10.00 | 3.00 | 12.00 | 7.65 | 1.68 | 13.22 | .81 |
| Dealing with problems | 10.00 | 9.00 | 1.00 | 10.00 | 6.93 | 1.75 | 18.28 | .81 |
| Coping with anxiety | 9.00 | 8.00 | 1.00 | 9.00 | 6.14 | 1.51 | 11.95 | .81 |
| Professional help-seeking | 9.00 | 8.00 | 2.00 | 9.00 | 7.30 | 1.28 | 49.37 | .61 |
| *Total score* | 79.00 | 69.00 | 18.00 | 72.00 | 52.21 | 7.59 | 0.42 | .97 |
| *Revised total score*[b] | 73.00 | 64.00 | 15.00 | 67.00 | 47.09 | 7.44 | 0.42 | .97 |

Notes.

[a]Lowest possible score for all (sub)scales is zero. Highest possible score is equal to the number of items in each (sub)scale.

[b]Revised total score is the final total score, based on revisions discussed below. This score excludes all items from the *relationships with others* subscale, which was removed from the final version of the scale.

appropriate professional help for their teenager if needed. Other items within this subscale had concordance rates between 45.9% and 83.1%.

## Reliability of the scale

Agreement coefficients for the subscales and the total score are presented in Table 4. Agreement coefficients were above the recommended .75 level for seven subscales, and was very high (.97) for the total score. Two subscales had lower reliability: *relationships with others* (.68); and *professional help-seeking* (.61). Reliability was therefore deemed to be high for the total score, acceptable to high for seven of the subscales, and questionable for two subscales.

Test-retest reliability was calculated for a smaller subsample of 175 participants in the waitlist control group of one of the RCTs, based on data from baseline to 1-month follow-up assessments. Test-retest reliability was deemed to be acceptable for the total score, $r = .76$, $p < .001$. Subscale test-retest reliability ranged from .29 to .72 ($p < .001$), with five of the nine subscales having test-retest reliability between .59 and .72 (see Table S1 for individual subscale test-retest reliability).

## Correlations between subscale scores and total score

We examined the correlation matrix of subscale scores and the total score (see Table 5). With the exception of the *relationships with others* subscale, all subscales had significant moderate to high correlations with the total score, as predicted. Also as hypothesised, most subscales had small to moderate correlations with the other subscales, reflecting the unique skills assessed within each domain. The *relationships with others* subscale had the lowest correlations with other subscales.
**Table 5  Pearson's Correlations between PRADAS Subscale Scores and PRADAS Total Score ($N =$ 711).**

|  | 1 Relationship | 2 | 3 | 4 | 5 | 6 | 7 | 8 | 9 |
|---|---|---|---|---|---|---|---|---|---|
| 2. Involvement | .35*** | – | | | | | | | |
| 3. Relationships with others | .09* | .21*** | – | | | | | | |
| 4. Family rules | .31*** | .31*** | .03 | – | | | | | |
| 5. Home environment | .40*** | .24*** | .08* | .28*** | – | | | | |
| 6. Health habits | .19*** | .22*** | .00 | .23*** | .18*** | – | | | |
| 7. Dealing with problems | .37*** | .32*** | .06 | .44*** | .33*** | .30*** | – | | |
| 8. Coping with anxiety | .29*** | .26*** | .12** | .31*** | .29*** | .19*** | .49*** | – | |
| 9. Professional help-seeking | .21*** | .20*** | .07 | .20*** | .21*** | .27*** | .28*** | .25*** | – |
| *Total score* | .62*** | .58*** | .23*** | .64*** | .61*** | .53*** | .73*** | .64*** | .51*** |
| *Revised total score*[a] | .62*** | .57*** | .13*** | .65*** | .61*** | .54*** | .74*** | .64*** | .51*** |

**Notes.**

[a] Revised total score is the final total score, based on revisions discussed below. This score excludes all items from the *relationships with others* subscale, which was removed from the final version of the scale.

*$p < .05$.

**$p < .01$.

***$p < .001$.

## Refinement of the scale based on analyses

Based on the above analyses, we chose to remove the *relationships with others* subscale from the final version of the scale for the following reasons: (1) there was a ceiling effect for the subscale score, attributable to ceiling effects for four of the six items; (2) questionable reliability (agreement coefficient = .68); and (3) this section had a low correlation with the total score ($r = .23$; in contrast to all other subscales which had correlations $\geq .51$) as well as low correlations with the other subscales (*rs* .00 to .21). After removing the 6-item *relationships with others* subscale, the *agreement coefficient* for the total score was unchanged (.97) and the test-retest reliability of the total score increased slightly (from .76 to .78).

Although the *professional help-seeking* subscale also had ceiling effects for three out of nine items, this may be at least partially attributable to the demographic characteristics of our sample, rather than the items. Most parents were highly educated, and had already taken a step towards seeking professional help by enrolling in an RCT aimed at reducing risk of depression and anxiety in their adolescent. Moreover, given that many parents and adolescents in our sample had a history of mental health issues, it is likely that these parents had greater knowledge of mental health support services than parents who have never had reason to access these. Further, the subscale total score did not have a ceiling effect, and the subscale had a moderate correlation with the total score. We therefore chose to retain this subscale, despite its lower reliability.

All subsequent analyses are based on the revised total score calculated without the items comprising the *relationships with others* subscale.

## Convergent validity

To examine convergent validity, correlations between the total and subscale scores of the PRADAS and the IPPA and EAC were examined (see Table 6). We predicted positive

**Table 6 Correlations between PRADAS revised total score and scores on the IPPA and EAC.**

| | N | Pearson's correlation with PRADAS revised total score |
|---|---|---|
| IPPA total score | 709 | .61 |
| IPPA trust subscale | 709 | .56 |
| IPPA communication subscale | 709 | .61 |
| IPPA alienation subscale | 709 | .42 |
| EAC reward global subscale | 704 | .59 |
| EAC punish global subscale | 704 | −.49 |
| EAC neglect global subscale | 704 | −.56 |
| EAC override global subscale | 704 | −.32 |
| EAC magnify global subscale | 704 | −.36 |

Notes.

IPPA, Inventory of Parent Peer Attachment, parent-report; EAC, Emotions as a Child Scale.

$ps < .001$ for all correlations.

correlations with all IPPA scores, as the items from the *alienation* subscale were already reverse scored. Four of the five EAC subscales (punish, neglect, override, magnify) assess undesirable parenting practices and were not reverse scored, hence we predicted our measure to correlate negatively with these subscales and positively with the *reward* subscale. In support of the convergent validity of the PRADAS, there was a moderate to strong positive correlation between the PRADAS total score and the IPPA total score ($r = .61$, $p < .001$). The total score also had moderate to strong correlations in the predicted directions with the trust and communication subscales of the IPPA ($rs = .56$ and $.61$, $ps < .001$), as well as three of the five EAC global subscales (reward, $r = .59$; punish, $r = −.49$; neglect $r = −.56$; all $ps < .001$). These subscales each have some overlapping content with our scale, hence the strength of the correlations are appropriate to the degree of overlap of constructs assessed. There were also small to moderate correlations in the predicted direction with the IPPA alienation subscale ($r = .43$, $p < .001$), as well as the EAC override and magnify subscales ($rs = −.32$ and $−.36$, $ps < .001$).

Examination of the correlations between the PRADAS subscale scores and subscales of the IPPA and EAC also revealed small to moderate correlations in the predicted directions. For example, the *parent–child relationship* subscale had a correlation of .51 ($p < .001$) with the IPPA total score and trust subscale, and lower correlations with the communication and alienation subscales ($rs = .45$ and $.35$, $ps < .001$), as well as the EAC reward, punish, and neglect subscales ($rs = −.42$ to $.46$, all $ps < .001$). Likewise, the *involvement*, *dealing with problems*, and *coping with anxiety* subscales all had small to moderate correlations in the predicted directions with various subscales of the IPPA and EAC (see Table S2 for subscale correlations). Two subscales of our measure assessed content not measured by the IPPA or the EAC: *health habits* and *professional help-seeking*. As expected, these subscales had the lowest correlations with all subscales of the IPPA and EAC (all $|r| < .32$; see Table S2). Overall, we interpreted these results as support for the convergent validity of the PRADAS.

**Table 7  Correlations between PRADAS revised total score, parent- and child-report SCAS and SMFQ scores, and parent and child characteristics.**

| | N | Pearson's correlation with PRADAS revised total score | p |
|---|---|---|---|
| Parent-report SCAS | 708 | −.13 | <.001 |
| Parent-report SMFQ | 708 | −.21 | <.001 |
| Child-report SCAS | 660 | −.08 | .044 |
| Child-report SMFQ | 658 | −.10 | .011 |
| Parent male gender | 711 | −.13 | <.001 |
| Child male gender | 711 | .04 | .355 |
| Parent age | 711 | .06 | .086 |
| Child age | 711 | −.05 | .187 |
| Parent education level | 711 | .11 | .004 |
| Parental concern re: child's risk of depression | 707 | −.14 | <.001 |
| Parent concern re: child's risk of anxiety | 708 | −.11 | .005 |
| Child current mental health problem | 698 | −.01 | .722 |
| Child history of mental health diagnosis | 677 | −.02 | .694 |
| Parent history or current mental health problem | 706 | −.05 | .224 |

**Notes.**

SCAS, Spence Children's Anxiety Scale; SMFQ, Short Mood and Feelings Questionnaire.

Parent and child gender coded as: 0, female; 1, male.

Parent education level coded on an 6-point scale following Australian national education standards (higher values represent higher levels of education; *Australian Qualifications Framework Council, 2013*); parental concern about child's risk of depression and anxiety coded from 1 = not at all concerned to 4 = very much concerned; child and parent mental health characteristics coded as 0 = no history/current mental health diagnosis (including parent-reported potential undiagnosed problem), 1 = history/current mental health diagnosis.

## Correlates of guidelines concordance
### Associations with adolescent symptoms

To examine the associations between parental concordance with the Guidelines and adolescent symptoms of anxiety and depression, we conducted bivariate correlations between the PRADAS revised total score and the parent- and child-report versions of the SCAS and SMFQ (see Table 7). There were small significant correlations in the predicted direction for both parent-report symptom measures, with higher parental concordance with the Guidelines associated with lower parent-reported adolescent symptoms. Correlations were very low, yet significant, for the child-report measures. The correlations between parent- and child-report symptom measures were moderate (SCAS: $r = .45$, $p < .001$; SMFQ: $r = .47$, $p < .001$).

To further explore the relationship between PRADAS scores and adolescent symptoms, we conducted *post hoc* exploratory analyses of participants above and below clinical cut-offs on the SCAS and SMFQ (see File S2 for details). Binary logistic regressions were performed to assess the ability of the PRADAS to predict elevated symptom status (i.e., above or below clinical cut-off) on each of the measures. The logistic regression models were statistically significant for the SCAS-P ($\chi^2[1] = 12.77$ $p < .001$, 77.5% correctly classified), SMFQ-P ($\chi^2[1] = 21.88$ $p < .001$, 77.4% correctly classified), and SMFQ-C ($\chi^2[1] = 7.41$ $p = .006$, 69.8% correctly classified), but not for the SCAS-C ($\chi^2[1] = 3.50$ $p = .061$, 85.5% correctly

classified). Effect sizes were small (Cohen's $ds = 0.15$ to $0.31$). Results of these analyses are presented in File S2, Table S3.

### Associations with participant characteristics

Finally, we explored the correlations between parent and child characteristics (gender, age, mental health history, parental concern about child's risk of depression/anxiety) and overall Guidelines concordance. As shown in Table 7, there were significant small positive correlations between overall Guidelines concordance and parent female gender, higher parent education level, and lower parental concern about child's risk of depression or anxiety. Parent or child age, child gender, and parent or child history of mental health diagnosis were not correlated with Guidelines concordance.

## DISCUSSION

The primary aim of this study was to validate the newly developed *Parenting to Reduce Adolescent Depression and Anxiety Scale* as a criterion-referenced measure of concordance with a set of evidence-based guidelines, which recommend parenting strategies for the prevention of adolescent depression and anxiety. To do this, we examined the reliability and convergent validity of the measure. Our second aim was to examine levels of concordance with the Guidelines in a large sample of Australian parents of adolescents aged 12 to 15. Finally, we examined the correlates of concordance with the Guidelines in this sample.

Our results provide preliminary support for the reliability and validity of the PRADAS. We removed one subscale from the final version of the scale, due to a ceiling effect, questionable reliability, and low correlations with the total score and other subscale scores. The final version of the scale therefore comprises 73 items, which are summed to form eight subscale scores and a total score. The total score and seven of the eight subscales had high reliability, as measured by the agreement coefficient. One-month test-retest reliability of the total score was acceptable to high, and varied for the subscale scores. Subscale scores had moderate to high correlations with the total score, indicating that concordance in one area of the Guidelines is associated with greater overall concordance. Most subscales had small to moderate correlations with the other subscales, suggesting that while there is some association between domains, each subscale reflects a set of unique parenting skills. The total and subscale scores correlated in the predicted direction and strength with the IPPA and EAC, supporting the convergent validity of the scale.

Based on our findings, we recommend that interpretation of scores on the PRADAS be primarily based on the total score, which we found to be the most reliable and valid score. This score gives an overall level of concordance with recommendations in the Guidelines, indicating the degree to which a parent's current parenting practices align with the highest quality evidence regarding parenting risk and protective factors for adolescent depression and anxiety. The eight subscale scores may also provide valuable information, particularly if specific domains of parenting skills are of interest. The subscales scores should, however, be interpreted in light of the findings of their lower reliability. This is particularly the case for the *professional help-seeking* subscale, which had questionable reliability. Results regarding this subscale may have been influenced by our sample demographics, as parents were highly

educated and had self-selected to participate in an RCT of an online preventive parenting intervention. Further, parents reported that they and their adolescents had slightly higher rates of current and past mental health diagnoses than have been reported in national prevalence studies (e.g., *Lawrence et al., 2015*; *Slade et al., 2009*). While our findings are based on parent-report of diagnosis, rather than rigorous assessment measures, it is possible that parents chose to participate in the research partially due to their own and/or their child's history of mental health issues. Future research with the scale would benefit from collecting data from participants with varying levels of education and mental health history.

## Parental concordance with the Guidelines

We examined levels of parental concordance with established Guidelines in an Australian community sample of parents of adolescents aged 12 to 15 years. Overall, parental concordance with the Guidelines was relatively low, with less than 20% of parents deemed concordant in six of the eight domains. This is unsurprising given that we intentionally set high cut-off scores for concordance. In contrast, the *professional help-seeking* domain had the highest concordance rate, at 49%. As discussed, this may be a result of our sample characteristics. Alternatively, parents may have scored highly in this domain due to the hypothetical nature of the questions. The items assess how likely a parent would be to respond in certain ways if they noticed a persistent change in their adolescent's mood or behaviour, as well as how likely they would be to seek professional help if needed. It is possible that parents may overestimate their likelihood to take particular actions when posed in hypothetical scenarios. As posited by theories such as the Theory of Planned Behaviour (*Ajzen, 1991*), and confirmed in numerous empirical studies (e.g., *Armitage & Conner, 2001*; *McEachan et al., 2011*; *Sheeran, 2002*), intentions do not always translate into action.

Concordance was lowest (4% of parents) for the *family rules* subscale, which assesses the establishment and consistency of rules and consequences. Closer examination of item responses revealed that less than half (47%) of parents endorsed having set specific, defined rules for their teenager's behaviour in most areas. Similarly, only 27% of parents answered that they had set specific, defined consequences for breaking the rules, and 16% said that their teenager had been involved in developing most of the rules. Promisingly, most parents (86%) stated that they adapt the family rules to their teenager's maturity and responsibility, and 80% said that they speak with their teenager about why their behaviour is not acceptable when enforcing a consequence. These findings are consistent with findings from a similar measure designed to assess parental concordance with guidelines for the prevention of adolescent alcohol misuse (*Yap, Jorm & Lubman, 2015*). This measure also assessed consistency of family rules, and found the *family rules* subscale to have the lowest concordance rate (7.2%). Together, these findings highlight an important target for future preventive programs, particularly given that parental disciplinary practices are associated with risk for both internalising and externalising disorders (*Berg-Nielsen, Vikan & Dahl, 2002*; *Gryczkowski, Jordan & Mercer, 2010*; *Yap et al., 2014a*).

The remaining six domains had concordance rates between approximately 10% and 20% of parents. Although this small percentage of parents scored above the cut-off score for concordance with the entire domain, it is promising to note that mean subscale scores

were close to or above the mid-range of possible scores. In other words, most parents answered within the concordant range for at least half of the individual items comprising each subscale, although they did not reach the threshold set for mastery of the skill area. This was also the case for the total score, which had a mean of 47 out of a possible 73. This means that while very few parents were considered concordant according to the cut-off scores set for "mastery" of each domain, on average parents were adhering to over half of the recommendations in the Guidelines. It is therefore plausible that many parents, particularly those who scored close to the cut-off, may move into the concordant range with intervention, where parents are supported to make changes in these parenting skill areas.

It could be argued that the designation of cut-off scores for concordance was somewhat arbitrary, and that the cut-off scores could be set lower given the low concordance rates obtained in our sample. As discussed, we chose high cut-off scores that indicate close-to-absolute-concordance with the Guidelines' recommendations. This was based on the expertise of the developing authors, the strong evidence base for the parenting factors assessed, and the purpose for which the scale was initially developed. The optimal cut-off scores for a criterion-referenced test such as this will depend on individual applications. For example, high cut-off scores may be appropriate when the goal is to evaluate intervention effects or to identify areas for improvement, whereas lower cut-off scores may be appropriate if attempting to determine a "bare minimum" level of competency. In the case of the PRADAS, we aimed to develop a measure that could be used to identify domains of parenting that could be improved by parents so they become more concordant with the Guidelines. From a public health perspective, prevention approaches that attempt to shift all individuals along the continuum of risk are likely to have greater effects at a population level than approaches only targeting those at highest risk (in this case, parents with the lowest PRADAS scores; e.g., see seminal work by *Rose, 1993*). To explore this, we examined scatterplots of PRADAS scores and scores on the symptoms measures with locally weighted smoothing (LOESS) lines fitted. These curves indicated that, even at the higher end of PRADAS scores, adolescent symptoms were lower, supporting the value of encouraging all parents (even those with moderate and high scores) to increase their concordance with the Guidelines. Finally, we conducted additional exploratory analyses with lower cut-off scores (one score less than the original cut-offs for all subscales). With these lower cut-off scores, reliability of the eight subscales and the total score reduced. Together, these findings support the use of high cut-off scores for the current purpose of the PRADAS. It should also be noted that the cut-off scores for concordance are one of two methods of interpreting PRADAS scores. The continuous scores are also calculated (i.e., total score out of 73, or subscale scores), and these can be interpreted alone or with reference to the subscale cut-off score. In many settings, the continuous scores are likely to be of most interest, as they allow interpretation of an individual's score in comparison to the range of possible scores. This may be of particular value in clinical contexts, as well as to track change over time.

## Correlates of parental concordance with the guidelines

We examined correlations between parental concordance with the Guidelines and parent- and child-report measures of adolescent depression and anxiety symptoms, as well as participant characteristics. We found small but statistically significant correlations between parental Guidelines concordance and adolescent symptoms. The correlations were slightly larger for parent-reported symptoms compared to child-reported symptoms, suggesting that parents who rate themselves as more concordant with the Guidelines also report their adolescents to have fewer symptoms of depression and anxiety. Additional exploratory logistic regression analyses revealed a small effect of the PRADAS in predicting current symptom elevation status (i.e., above or below clinical cut-off) for three of the four symptom measures (SCAS-P, SMFQ-P, SMFQ-C). To follow up on these findings, we also compared mean PRADAS scores for participants above and below clinical cut-offs. In line with the logistic regression results, PRADAS total scores were significantly lower for participants who scored above the clinical cut-off on the SCAS-P, SMFQ-P and SMFQ-C (see Table S4). Together, these findings suggest that there is a relationship between PRADAS scores and current clinically elevated symptom status, albeit a modest one. Similarly, there was a small, significant correlation between Guidelines concordance and parental concern about their child's risk of developing both depression and anxiety. Parents with higher concordance were less concerned about their child's mental health. The findings regarding adolescent symptoms should be interpreted with caution, due to the small effects obtained and design of the study. The cross-sectional analyses limits our ability to assess the theorised longer-term risk factors that the PRADAS is designed to assess. That is, the factors assessed by the scale are expected to increase or decrease risk of depression or anxiety problems developing throughout adolescence. More rigorous assessment of the relationship between PRADAS scores and adolescent symptoms requires prospectively collected data, which would more accurately assess the predictive validity of the scale. Additionally, the symptom measures could be considered "state" measures (i.e., assessing current/recent symptoms), whereas the PRADAS may be more likely to capture "trait" level parenting practices. A cross-sectional design may not adequately capture the relationship between these variables. Also, we intentionally recruited a community sample, yielding mean scores on the symptom measures that were below clinical cut-offs. It would be of interest to examine the associations between PRADAS scores and symptom levels in a clinical population. Finally, meta-analyses of the associations between parenting and child/adolescent symptoms have also reported small effect sizes (e.g., *McLeod, Weisz & Wood, 2007*; *McLeod, Wood & Weisz, 2007*; *Yap et al., 2014a*; *Yap et al., 2014b*). Given these points, it is unsurprising that the associations between PRADAS scores and symptom measures were small.

Regarding demographic characteristics, there were small significant correlations of parent female gender and higher levels of education with higher Guidelines concordance. This should be interpreted in light of the over-representation of female, highly-educated parents in our sample. This is in line with previous findings of higher rates of mental disorders among children of parents with lower education (e.g., *Lawrence et al., 2015*; *Meltzer et al., 2003*; *Merikangas et al., 2010*), as well as an association between parental

education level and treatment outcomes in parenting intervention studies (e.g., *Reyno & McGrath, 2006*). Additionally, in a survey of Australian parents' beliefs about the role of parenting in the prevention of depression in young people, parents with tertiary-level education were more likely to hold beliefs consistent with the evidence regarding parental over-control as a risk factor for depression (*Yap & Jorm, 2012*). The current findings contribute further evidence highlighting the need to consider socio-demographic factors, such as parental education level, when considering strategies to target in preventive parenting programs.

## Limitations and future directions

The current findings should be interpreted in the context of study limitations. Our sample had an overrepresentation of highly-educated mothers from intact families, limiting the generalisability of findings to fathers, those with lower levels of education, and different family situations. Nonetheless, while our sample may not be representative of the general population, it is likely representative of parents who self-select to participate in online parenting programs. Our sample characteristics are similar to those often reported in online parenting intervention studies (e.g., *Dittman et al., 2014*; *Enebrink et al., 2012*; *March, Spence & Donovan, 2009*; *Sanders, Baker & Turner, 2012*). Importantly, given the correlation found between education and parental Guidelines concordance, concordance rates in our sample may be higher than would be found in less educated populations. It is an important, yet challenging, direction for future research to recruit more diverse samples of parents into research of this nature.

The current study is the first psychometric evaluation of the newly developed PRADAS. Use of the PRADAS in future research will allow for further refinement of the measure. This would be particularly valuable to improve the utility of the subscale scores, so as to provide eight short scales that can be used with confidence to assess the various domains of parenting covered in the Guidelines. The current form of the scale is quite long, with 73 items, and mean completion time just under 18 min ($M = 17.76$, $SD = 12.93$). It would therefore be valuable to develop a short-form of the scale, which could have greater utility in clinical settings. Additionally, development of an adolescent-report version of the PRADAS would allow cross-validation between parent- and adolescent-report, which is not captured in the current version of the scale.

The associations between PRADAS scores and adolescent symptoms were small. The cross-sectional nature of the study limits our ability to assess the prospective predictive validity of the scale. We are also unable to ascertain concurrent validity based on diagnostic status, which cannot be determined from symptom measures such as those used in the used in the current study. However, it should be noted that the PRADAS is designed to assess parenting factors which evidence and expert consensus suggest would influence risk of adolescent depression and anxiety; the measure in itself does not purport to predict future risk. Results of the PRADAS should therefore not be interpreted as indicative of risk for the adolescent.

To our knowledge, the PRADAS is the first assessment measure to comprehensively assess the range of parenting risk and protective factors shown to influence the development

of depression and anxiety disorders in adolescents. This is a valuable addition to the growing body of research investigating prevention of depression and anxiety in young people. It has potential for use in both research and clinical settings. The scale may serve as an assessment measure to tailor and evaluate preventive interventions targeting parenting factors, such as in the RCTs mentioned above. For investigators interested in specific domains of parenting, the option of eight subscales assessing different risk and protective factors may be of particular interest. Clinically, the PRADAS could be used to identify individual parents' strengths and areas for improvement, which may be valuable to clinicians working with parents. Future studies could examine parental Guidelines concordance in clinical versus community samples, including parents of adolescents receiving treatment for depression or anxiety. It would be of interest to ascertain whether the same parenting factors are relevant to treatment or relapse prevention outcomes in a clinical population. If the PRADAS was found to contribute valuable information in this population, it could be used to tailor interventions for parents of adolescents experiencing depression or anxiety disorders.

## CONCLUSIONS

This paper presented the development and validation of the *Parenting to Reduce Adolescent Depression and Anxiety Scale*—a self-report, criterion-referenced measure that assesses parental concordance with a set of evidence-based parenting guidelines for the prevention of adolescent depression and anxiety disorders. Our findings provide preliminary support for the reliability and validity of the scale as a measure of parental concordance with the Guidelines, which reflect high-quality research evidence and expert consensus regarding parenting risk and protective factors for adolescent depression and anxiety. Its ease of use and coverage of a broad range of parenting factors make the PRADAS a valuable measure for both clinical and research settings. It can be used to identify individual parents' strengths and areas for improvement, or as an evaluative measure in studies of preventive interventions targeting modifiable parenting factors for adolescent depression and anxiety disorders.

## ACKNOWLEDGEMENTS

The authors would like to acknowledge the RCT project managers Mrs Jennifer Hanson-Peterson, Dr Shireen Mahtani, Ms Claire Nicolas, and Ms Jacqueline Green, as well as the team of research assistants who assisted with data collection. We would like to thank the parents in our reference group for their input into the development of the PRADAS. We also acknowledge *beyondblue* for their partnership in developing and disseminating the Guidelines, and Mental Health First Aid Australia and the schools who assisted with recruitment for the two RCTs.

### Funding

This work was supported by: Monash University Faculty of Medicine, Nursing and Health Sciences Faculty Strategic Grant Scheme funding (SGS15-0149); Australian Rotary

Health Research Grant; and the Windermere Foundation Doctoral Scholarship in Health (Allied Health, 2016). Mairead C. Cardamone-Breen received an Australian Government Research Training Program Scholarship. Marie B.H. Yap received salary support from a National Health and Medical Research Council (NHMRC) Career Development Fellowship (APP1061744). Anthony F. Jorm received salary support from an NHMRC Senior Principal Research Fellowship (APP1059785). The funders had no role in study design, data collection and analysis, decision to publish, or preparation of the manuscript.

### Grant Disclosures

The following grant information was disclosed by the authors:
Monash University Faculty of Medicine, Nursing and Health Sciences Faculty Strategic Grant Scheme: SGS15-0149.
Australian Rotary Health Research Grant.
Windermere Foundation Doctoral Scholarship in Health (Allied Health, 2016).
Australian Government Research Training Program Scholarship.
National Health and Medical Research Council (NHMRC) Career Development Fellowship: APP1061744.
NHMRC Senior Principal Research Fellowship: APP1059785.

### Competing Interests

Anthony F. Jorm is an Academic Editor for PeerJ.

### Author Contributions

- Mairead C. Cardamone-Breen conceived and designed the experiments, performed the experiments, analyzed the data, contributed reagents/materials/analysis tools, wrote the paper, prepared figures and/or tables, reviewed drafts of the paper, contributed to the development of the scale.
- Anthony F. Jorm conceived and designed the experiments, contributed reagents/materials/analysis tools, reviewed drafts of the paper, contributed to the development of the scale.
- Katherine A. Lawrence conceived and designed the experiments, reviewed drafts of the paper, contributed to the development of the scale.
- Andrew J. Mackinnon analyzed the data, contributed reagents/materials/analysis tools, reviewed drafts of the paper.
- Marie B.H. Yap conceived and designed the experiments, performed the experiments, contributed reagents/materials/analysis tools, reviewed drafts of the paper, contributed to the development of the scale.

### Human Ethics

The following information was supplied relating to ethical approvals (i.e., approving body and any reference numbers):

Ethics approval was obtained from the Monash University Human Research Ethics Committee (approval numbers CF14/3886—2014002023 and CF14/3887—2014002024).

## Data Availability

Cardamone-Breen, Mairead; Yap, Marie BH (2017): Raw data for manuscript "The Parenting to Reduce Adolescent Depression and Anxiety Scale: Assessing parental concordance with parenting guidelines for the prevention of adolescent depression and anxiety disorders". PeerJ, accepted 29th August 2017. figshare. https://doi.org/10.4225/03/592b73c2e911c.

## Supplemental Information

Supplemental information for this article can be found online at http://dx.doi.org/10.7717/peerj.3825#supplemental-information.

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
