# Peer review of "The Parenting to Reduce Adolescent Depression and Anxiety Scale: Assessing parental concordance with parenting guidelines for the prevention of adolescent depression and anxiety disorders"

_PeerJ, doi:10.7717/peerj.3825_

## Round 0.1 · original submission · Major Revisions

· Academic Editor

Major Revisions

Thank you for submitting your article to PeerJ. I have now received three reviews and I would like to thank all the reviewers for their thoughtful assessments of the manuscript. The reviews are appended below. The reviewers have highlighted important issues that need to be addressed in any revision.

Please address all reviewer comments; however, I believe that the following comments warrant particular attention when revising your manuscript:

1) Both Reviewer 2 and Reviewer 3 had concerns about the factor analyses. These will need to be addressed in any revision. In particular, the comment regarding the choice to factor analyse subscale scores rather than individual items needs consideration.

2) Both Reviewer 1 and Reviewer 3 raised concerns over the choice of cut-off scores that indicate concordance with items/scales, and the fact that only a small proportion of parents were concordant with the guidelines. I agree with both reviewers that this is an important point to consider in your response.

3) As Reviewer 3 highlights, the associations between the new scale and parent and child reported emotional outcomes are very small. The fact that associations are “small but significant” is mentioned in the discussion; however, given that the scale is specifically designed to assess parenting factors that might prevent the development of mental health problems in adolescents, this finding needs much more discussion.

4) Finally, Reviewer 3 questioned the name of the measure, and I have to admit to also having had some misgivings as to whether “Parenting Resilient Adolescents” was the best label. I would encourage you to at least consider whether an alternative name might prevent potential confusion.

Thank you again for submitting your article to PeerJ. I hope the reviewers’ comments are helpful in revising your manuscript, and that the points above are useful in focusing your response.

Reviewer 1 ·

Basic reporting

The manuscript is extremely well-written, logical and easy to follow.

Experimental design

The aims of the project are clear, the sample size is impressive given it was collected through two RCTs. For the most part, any decision making processes with regard to design and analyses are well articulated and well justified.

The one clear exception is in choosing the cut-off scores that indicate concordance with each item and with each scale. I acknowledge the authors' position that only high scores can indicate mastery, but the choice of cut-off seems somewhat arbitrary. Is there not a more objective way to assess the ideal cut-off that would meaningfully distinguish the level of concordance that might increase/decrease risk of anxiety and depression in adolescents? An indication of specificity/sensitivity of the measures would be helpful in determining clinical utility. As currently scored only 'perfect' parents appear to be concordant with the guidelines, but 100% concordance may not be needed to produce resilience adolescents.

Validity of the findings

The conclusions are generally valid, however I feel a stronger argument could be made if there was an empirically-grounded method of establishing optimal cut-offs on the items and scales.

The limitations inherent in using cross-sectional data to assess relationships between parenting practices and mental health outcomes in adolescents could be explicitly addressed.

Reviewer 2 ·

Basic reporting

This manuscript provides a very rich description of the development of a measure to assess parental concordance with parenting guidelines aimed at the prevention of adolescent depression and anxiety disorders. I have read the manuscript with interest and would like to offer comments on how the manuscript can be improved.

Overall, the manuscript is very dense and overly long. It would be beneficial if the authors shortened and clarified the steps undertaken. There are a number of sections in which content could be streamlined i.e. methods, measures, results and discussion.

References and literature provided in the background section are for the most part sufficient and give a good overview.

P 4, ln 64 the authors state that some risk factors for depression are not easily amenable to preventive interventions and give child abuse and neglect as well as stressful life events an example while in line 67 they cite inter-parental conflict as a risk factor that may be more easily modifiable. I think this argument is slightly misleading for the following reasons: 1. Parenting interventions (such as the one this new measure might be used for) have been shown to decrease child abuse and neglect (i.e. Barlow et al, Gardner et al, Cluver et al, Saunders et al, Miller et al or for systematic reviews Mikton and Butchart 2009, Closkey et al 2011 or Knerr et al 2011). 2. Child abuse and neglect often co-occur with inter-parental conflict and parental aversiveness and thus it does not necessarily follow that one is easily modifiable while the other isn’t, particularly while few interventions are actually effective in reducing intimate partner violence.

The background could be strengthened by the inclusion of evidence from systematic reviews that show that family and parenting factors play a role in the prevention of depression and anxiety or the improvement of child mental health i.e Siegenthaler et al 2012 even if much of the parenting interventions have focused on child externalizing behavior

P 9 ln 85 typo risk of risk

Experimental design

The research definitely falls within the aims and scope of the journal and the research question is well defined, relevant and meaningful. It is also stated how the research fills an identified knowledge gap.

I have a few comments on the methodology chosen for this study which I think will strengthen the manuscript:

The methods section could be more streamlined. It currently provides a lot of information in a not so concise way which is confusing for the reader. Examples of this are provided below.

Please state additional information about consent procedures for both caregivers and children and protocols for privacy, confidentiality and abuse disclosure (you mention some of this below the measures section and it could be moved up here to improve text flow)

Should adolescents not also be asked the questions in the Parenting Resilient Adolescents Scale or should they be asked a sub-set of the questions to validate parental claims with regards to their parenting?

Why was the IPPA administered to parents and not to adolescents? Is it not a scale that was developed specifically to measure adolescent’s attachment to their parents? Has this measure been validated for use in parent populations?

IPPA measure – Cronbach’s alpha for some of the measures range from .68. It was always my understanding that .70 was considered acceptable and anything below that was poor – as overall your scales performed well I would report this one as the poor result that it is.

P 18. I think text flow could be improved if the procedure section here could be moved up to just under the participant section where it follows on from the ethics statement. Could authors also provide more information on how they dealt with adolescents with low literacy and disabilities?

Could the authors please clarify for all measures who the respondents are. Please also clarify where necessary if the measure has been validated/normed for both respondents or only one.

Missing data: please state how much missing data there was overall for the IPPA, EAC, SCAS and SMFQ

I do not understand why the authors chose to use subscale scores in their EFA rather than all of the items? Would it not be wise to examine whether the different items actually load onto the individual domains of the subscales as hypothesized? It would then be good to give alphas for the remaining sub-scales with the items that are actually loading onto them rather than the ones you hypothesize to load onto them. Alternatively, the authors could have conducted a CFA to examine model fit of the hypothesized dimensions prior to conducting scale or total scale scores.

In your results please state actual p values for your results whether they are deemed statistically significant or not

Validity of the findings

Based on my comments provided on the methodology of the study I fell that the authors exaggerate the validity and reliability of the measure. A study examining the construct validity of a new measure based on factor scores needs further recommendations as to the construct validity of the individual dimensions within the measure.

A short section on how this measure can contribute to research and practice would be nice

There are a few sections with repetition which could be streamlined i.e. p36 ln 644

Reviewer 3 ·

Basic reporting

The quality of expression is excellent throughout this manuscript. The literature review is clear, coherent, and thorough. The supportive tables are illustrative and used effectively, and the raw data shared.

Experimental design

The research question was well defined and relevant. I did question the name of the measure, as it is specific to prevention of mood/anxiety problems rather than resilience promotion. resilience is about wellness -- it is not merely the absence of illness or problematic emotional functioning. I would suggest changing the name in order to reflect an appreciation of how resilience indicates more than the absence of anxiety and depression in teens.

The authors used 2 samples, which were comparable on most demographics, and then collapsed these into a single sample. I did not have problems with this based on the information provided and felt it was well justified.

The methods were very well described and would permit replication

Validity of the findings

see below

Additional comments

Thank you for asking me to review this manuscript, “Parenting Resilient Adolescents Scale: Assessing parental concordance with parenting guidelines for the prevention of adolescent depression and anxiety disorders,” submitted for publication at PeerJ. This reports the development of the “Parenting Resilient Adolescents Scale (PRAS),” which is intended to assess parents’ engagement in parenting behaviours that prevent or protect their adolescents from developing anxiety and/or mood disorders. Two samples were collected, which were then summed together to form a larger sample of 711 parents. Factor analyses indicated a 1-factor solution. However, the authors justify their choice of adopting 9 subscales and then eliminating 1 for a total of 8 subscales based on their explicit goal of creating a criterion-referenced measure based on empirically-supported parenting practice guidelines for prevention of anxiety/mood problems in teens. Reliability was unacceptable for one of these subscales, and acceptable to high for the remaining subscales.

Criterion referenced scales can serve a needed function, as they can be used to judge the distance between an individual’s current and desired functioning, rather than the distance between an individual’s current and the population mean’s level of functioning.

There are many positive aspects to this paper, which I have pointed out in the above sections (basic reporting and experimental design).

Having said this, I have concerns about the utility of the scale that the authors provide.

First, this scale is specifically developed to assess parents’ engagement in parenting skills that prevent adolescents’ development of mood/anxiety problems. However, I am not convinced that the scale does this in the end. In particular, the correlations between the PRAS (considered as 1 scale total) and measures of parent- and child- reported emotional problems are very small (~-.1; indicating about 1% of shared variance). In fact, the relationship between the PRAS and parent education is on par (.11). Although there are similar relationships with parents’ concerns about risk of depression and anxiety (also ~-.1), the relationships with current or past mental health problems/diagnoses are non-significant (~.02).

Although this is interpreted as evidence for the validity of the PRAS, taken together with a lack of supportive factor analytic findings, it questions this interpretation.

I have some concerns about the utility of the scale. It was difficult to find the total number of items in the final scale, but adding up those listed in Table 2 for each of the subscales results in 69 items. In both research and clinical contexts, a scale of this length would be onerous for participants/clients, and thus would need an excellent justification for adopting.

Finally, I would suggest adopting lower cut-offs for considering whether or not parents are concordant with the guidelines. That only 20% are concordant in 6 or more of the subscales suggests that the vast majority of teens are at risk of developing a mood/anxiety disorder. Softer cut-offs might be more encouraging and realistic for families to meet.

---

## Round 0.2 · Minor Revisions

· Academic Editor

Minor Revisions

Thank you for your thorough response to the reviewers comments. I have now received reviews of the revised manuscript from all three of the original reviewers, and I would like to thank them again for their contribution.

As you can see, Reviewer 3 still had significant concerns about i) the cut-offs and ii) the size of the associations between PRADAS scores and adolescent mental health scores. I share these concerns; however, I believe that some additional revisions to the discussion, whilst not necessarily solving the underlying issues, could directly tackle these criticisms. Additionally, do the mental health measures have clinical cut-offs? If so, examining whether PRADAS scores are associated with probability of scoring in the clinical range may address the potential clinical utility of the scale (even if associations with total mental health scores are weak).

Thank you again for submitting your manuscript to PeerJ.

Reviewer 1 ·

Basic reporting

no comment

Experimental design

no comment

Validity of the findings

no comment

Additional comments

The authors have addressed my initial comments well.

Reviewer 2 ·

Basic reporting

The authors have responded well to all of my previous concerns around structure and references.

Experimental design

No comment

Validity of the findings

The validity of the findings would be strengthened if the authors could demonstrate that the PRADAS is sensitive enough to detect change in parenting. If the RCTs have been concluded it would be wonderful if this information could be included.

Additional comments

Other than the above the authors have addressed all of my concerns.

Reviewer 3 ·

Basic reporting

The general level of English used throughout is clear and unambiguous, and the literature cited is sufficient.

Experimental design

I am pleased to see the name change of the scale, which more closely and accurately depicts the authors' intentions of the content that they aim to capture.

Validity of the findings

no comment

Additional comments

I still have concerns about the utility of the sale that are not addressed in this version. Most notably, the length of the scale is too long to be included in most situations (whether research or clinical). The authors have acknowledged this problem and have added to the revised manuscript suggesting that this is a problem for future research to tackle; however, I would still suggest that this version of the manuscript, given the very sizeable participant database and that they authors are the composers of this scale, would be the ideal place to take on such a task.

I also have remaining concerns about the fact that only 20% of parents report that their parenting is within the guidelines of not placing their adolescent at risk for depression/anxiety. The authors indicate that the LOESS lines fitted for teens at the upper end of the PRADAS support the contention that those teens are still less symptomatic, and thus there are still benefits to be had in improving parenting even at the top end of the PRADAS possible scores. Given that the correlation between adolescents' anxiety/depression symptoms and the PRADAS is very low (~1% of shared variance), this argument of relationship is not compelling. Also, there is still the problem that 80% of the population (in accordance with the current figures) are placing their child at risk for something that only 15-20% of teens will develop. I understand this approach of placing a very high cutoff if there are other intentions of the scale -- e.g., if this scale was really designed as a general resilience scale (as opposed to one that helped to prevent or protect against depression/anxiety specifically), and there was evidence that 80% of teens are at risk for low resilience. In all, my concern that the majority of parents are seen on this scale to have inadequate parenting still stands.

Finally, the authors addressed reviewers' concerns that there are very small correlations between adolescent depression/anxiety and parenting on the PRADAS (indicating about 1% of shared variance) by stating that these correlations are only concurrent, not longitudinal, and thus cannot really capture the possible correspondence that exists between teens depression/anxiety development and the risk posed to them by their parents' parenting behaviours during this time period. Instead, it is suggested that longitudinal studies would better be able to address this concern. I agree that longitudinal studies are more appropriate to test this contention; and in fact, that these are even needed in order to provide validity information for a new scale that purports to assess future risk. However, this information is not available yet, and thus the only indication of validity is one that the authors themselves point out is less than optimal, and at correlations of ~.1, is less than highly supportive. This means that the potential concurrent/divergent validity of the PRADAS must be judged on the information provided, which is limited.

---

## Round 0.3 · accepted · Accept

· Academic Editor

Accept

Thank you for your revised submission and your responses to the reviewers' comments. I am pleased to accept your manuscript for publication in PeerJ.